# Extended Versus Intermittent Meropenem Infusion in the Treatment of Nosocomial Pneumonia: A Retrospective Single-Center Study

**DOI:** 10.3390/antibiotics12101542

**Published:** 2023-10-15

**Authors:** Dong-gon Hyun, Jarim Seo, Su Yeon Lee, Jee Hwan Ahn, Sang-Bum Hong, Chae-Man Lim, Younsuck Koh, Jin Won Huh

**Affiliations:** 1Department of Pulmonary and Critical Care Medicine, Asan Medical Centre, University of Ulsan College of Medicine, Seoul 05505, Republic of Korea; dghyun@amc.seoul.kr (D.-g.H.);; 2Department of Pharmacy, Asan Medical Centre, Seoul 05505, Republic of Korea

**Keywords:** meropenem, intensive care unit, pneumonia, ventilator-associated pneumonia, intravenous infusion

## Abstract

The efficacy of extended meropenem infusions in patients with nosocomial pneumonia is not well defined. Therefore, we compared the clinical outcomes of extended versus intermittent meropenem infusions in the treatment of nosocomial pneumonia. We performed a retrospective analysis of extended versus intermittent meropenem infusions in adult patients who had been treated for nosocomial pneumonia at a medical ICU between 1 May 2018 and 30 April 2020. The primary outcome was mortality at 14 days. Overall, 64 patients who underwent an extended infusion and 97 with an intermittent infusion were included in this study. At 14 days, 10 (15.6%) patients in the extended group and 22 (22.7%) in the intermittent group had died (adjusted hazard ratio (HR), 0.55; 95% confidence interval (CI): 0.23–1.31; *p* = 0.174). In the subgroup analysis, significant differences in mortality at day 14 were observed in patients following empirical treatment with meropenem (adjusted HR, 0.17; 95% CI: 0.03–0.96; *p* = 0.045) and in Gram-negative pathogens identified by blood or sputum cultures (adjusted HR, 0.01; 95% CI: 0.01–0.83; *p* = 0.033). Extended infusion of meropenem compared with intermittent infusion as a treatment option for nosocomial pneumonia may have a potential advantage in specific populations.

## 1. Introduction

Nosocomial pneumonia represents one of the most common hospital-acquired infections and is a global public health concern associated with high mortality rates [1,2,3]. Causative pathogens of nosocomial pneumonia include Gram-negative bacteria and an increasing number of antimicrobial-resistant pathogens, such as carbapenem-resistant *Acinetobacter baumannii* (CRAB) and carbapenem-resistant *Pseudomonas aeruginosa* (CRPA) [4,5,6]. The surge in drug resistance and the scarcity of new antibacterial drugs have caused efforts to be focused on optimizing antibiotic dosing [7,8]. The antibacterial activity of beta-lactam antibiotics depends on the duration of its concentration above the minimal inhibitory concentration (MIC) of the causative bacteria [9]. However, critically ill patients often undergo changes in antibiotic pharmacokinetic profiles, leading to insufficient drug concentrations [10,11,12]. Since inadequate antibiotic exposure reduces antibacterial susceptibility in patients with sepsis, the optimization of antibacterial concentrations is warranted [13,14]. Attaining the target concentration through extended or continuous infusion of beta-lactam antibiotics has been associated with lower mortality rates in patients with sepsis or severe infections [15,16,17].

Antibiotics administered intravenously must diffuse across the structurally complex blood–alveolar barrier into the lungs [18]. As a result, the intrapulmonary concentration, represented as the epithelial lining fluid (ELF), is substantially lower than the serum concentration [19]. To achieve a higher exposure in the ELF compared with intermittent infusions, the administering of an extended infusion (EI) of meropenem has been introduced for critically ill patients with nosocomial pneumonia [20,21,22]. However, the efficacy of meropenem EIs compared to nosocomial pneumonia remains controversial [23,24]. Therefore, we aimed to compare clinical outcomes between the EI of meropenem and intermittent infusion in patients with nosocomial pneumonia.

## 2. Results

We assessed 687 patients who had undergone meropenem treatments for inclusion (Figure 1); among them, 526 patients were excluded because they did not receive meropenem in the ICU for more than 48 h, were not diagnosed with nosocomial pneumonia, or had received meropenem treatment before enrollment. Thus, 161 patients—64 undergoing extended infusion and 97 undergoing intermittent infusion—were included in this study.

### 2.1. Baseline Characteristics

The baseline characteristics were generally balanced between the two groups (Table 1). The mean age of all the patients was 63.8 ± 12.9, and over two-thirds of all participants were male. Being immunocompromised was the most common condition amongst participants in the extended group (34.4%), while diabetes mellitus (32.0%) was the most common in the intermittent group. The SOFA score and PaO_2_/FiO_2_ ratio were similar between the two groups. A total of 26 (40.6%) and 38 (39.2%) patients in the extended and intermittent groups, respectively, were diagnosed with ventilator-associated pneumonia (VAP). Antibiotic use revealed no difference in the median duration between the two groups (9.0 days in the EI group vs. 7.0 days in the II group, *p* = 0.165). Most patients (66.4%) discontinued meropenem treatment during their ICU stay due to the de-escalation or cessation of antibiotics. There were no differences between the previous use of antibiotics and the concomitant use of antibiotics across the two groups. Before meropenem treatment, the most commonly prescribed class of antibiotics was beta-lactam, followed by quinolone and glycopeptide. In addition, the two groups showed no significant difference in laboratory variables or organ support proportions at the baseline.

### 2.2. Microbiological Data

The microbiological profiles showed a similar pattern across the two groups (Table 2). Here, 8 (12.5%) of the 64 patients in the EI group and 5 (5.2%) of the 97 patients in the II group had bacteremia documented from blood cultures (*p* = 0.094). Furthermore, one patient (12.5%) in the EI group and five patients (100.0%) in the II group had the same bacteria cultured from both blood and sputum, while three patients (37.5%) in the EI group had different bacteria at each culture site. The identification of pathogens in the sputum culture totaled 37 (57.8%) and 59 (60.8%) in the EI and II groups, respectively (*p* = 0.744). One patient (2.7%) in the EI group had both *Staphylococcus aureus* and *Klebsiella pneumoniae*. Two patients (3.4%) in the II group had *Stenotrophomonas maltophilia* alongside other pathogens, such as *Klebsiella pneumonia* and *Pseudomonas aeruginosa.* From the multi-drug resistance (MDR) pathogens in the sputum culture, CRAB was isolated in 7 (10.9%) specimens taken from the EI group and in 21 (21.6%) from the II group (*p* = 0.092). Each patient in the EI (2.7%) and II groups (1.7%) possessed *Klebsiella pneumonia* and *Stenotrophomonas maltophilia*, alongside CRAB. Six patients (9.4%) in the EI group and four (4.1%) in the II group had CRPA (*p* = 0197). Only four patients (4.1%) in the II group had Carbapenemase-producing Enterobacteriaceae (CPE; *p* = 0.152). No difference was observed for any other pathogens between the two groups (*p* = 0.104).

### 2.3. Outcomes

Mortality occurred at 14 days for 10 patients (15.6%) in the EI group and for 22 (22.7%, *p* = 0.272) in the II group (Table 3). The clinical and microbiological outcomes were generally similar between both groups. There was no difference in the proportion of CRP level improvement between the two groups (65.1% in the EI group vs. 64.2% in the II group, *p* = 1.000). Further, 24.4% of the patients in the EI group presented with carbapenem resistance, while 26.9% possessed it in the II group (*p* = 0.829). Even after adjusting for covariables, including solid cancer, chronic liver disease, SOFA scores, septic shock, VAP, previous use of quinolone and glycopeptide, and bacteremia (Appendix A), there was no significant difference between the two groups (adjusted HR, 0.55; 95% CI: 0.23–1.31, *p* = 0.174; Figure 2). In addition, the two groups had similar rates of ventilator liberation (adjusted HR for PaO_2_/FiO_2_ ratio, use of colistin, and lactate level, 0.93; 95% CI: 0.51–1.68, *p* = 0.80) and rate of discharge from the ICU (adjusted HR for solid cancer, hematologic malignancy, respiratory failure, mechanical ventilator, and use of colistin, previous use of quinolone, 0.77; 95% CI 0.44–1.35, *p* = 0.36) at day 14 (Appendix A).

### 2.4. Subgroup Analysis

In a subgroup analysis among both the empirical treatment groups, the mortality rates at day 14 were lower in patients with EI compared with the II patients (adjusted HR for solid cancer, VAP, SOFA score, and previous use of penicillin and quinolone, 0.17; 95% CI: 0.03–0.96, *p* = 0.045; Appendix A) and the culture-proven group (adjusted HR for liver cirrhosis, bacteremia, SOFA score, septic shock, Gram-positive pathogen, and previous use of quinolone, 0.01; 95% CI: 0.01–0.83, *p* = 0.033; Figure 3 and Appendix A). Moreover, lower morbidity at 14 days was observed among patients with CRE—including CRAB and CRPA—in the EI group (0.0%) compared to those in the II group (25.0%), although these differences were not statistically significant (*p* = 0.056; Appendix A). Additionally, the baseline characteristics, absolute standardized mean differences, and results of the primary outcome in the propensity-matched cohort and the subgroup of patients with concurrent bacteremia are shown in Appendix A, respectively.

## 3. Discussion

This single-center study was retrospectively conducted to compare clinical outcomes between patients undergoing EI and II of meropenem. Our results showed that EI provided no additional survival benefits compared to II in the treatment of nosocomial pneumonia, even after controlling for any covariates. The incidence of carbapenem resistance following EI was similar to that after II. In the subgroups, EI of meropenem resulted in lower 14-day mortality rates than in patients receiving II, who were empirically treated for nosocomial pneumonia or had culture-proven pathogens. Although there were no differences in the clinical outcomes according to the type of infusion across all patients, EI of meropenem may confer a survival advantage over II in a specific population of patients with nosocomial pneumonia.

Several novel antibiotics, including ceftazidime–avibactam, ceftolozane–tazobactam, and cefiderocol, have been developed to target nosocomial pneumonia caused by Gram-negative pathogens [25]. Although these drugs have shown an efficacious and tolerable profile for the treatment of nosocomial pneumonia, their clinical outcomes are not superior to those of meropenem [26,27,28]. In addition, reports of resistance to novel beta-lactam/beta-lactamase inhibitor combination agents, such as ceftolozane–tazobactam, have recently emerged [29,30]. Several efforts, including antimicrobial stewardship programs, have been introduced to reverse the large burden associated with antimicrobial resistance and to preserve antimicrobial armamentariums in the ICU [8]. Optimizing the appropriate initial antibiotic therapy for critically ill patients is a key component of these interventions [31]. This strategy improves patient outcomes and reduces the emergence of resistance, while also avoiding the use of unnecessary antibiotics [7]. Although no statistically significant differences were observed, patients who received EI also had a lower mortality rate at 14 days than those in the II group, according to our subgroup analysis of patients with CRE. Therefore, attempts to optimize antibiotic concentrations in critically ill patients in the ICU are important.

The clinical outcomes of our study were consistent with those of previous studies. Indeed, a randomized-controlled, single-center study in China presented 28-day survival rates of 18.4% and 40.0% in patients with hospital-acquired pneumonia undergoing treatment with 1 g meropenem via 3 h infusions and 30 min infusions, respectively [32]. In another study comparing the efficacy of ceftolozane–tazobactam with meropenem for the treatment of nosocomial pneumonia, a mortality rate of 25.3% was observed at 28 days in patients who received meropenem (1 g by 30 min infusion every 8 h), which was similar to the II group in our results (29.9%) [27]. Recently, there was a randomized clinical trial for patients with sepsis or septic shock who received 1 g meropenem by either continuous or intermittent infusion [33]. No statistically significant difference was observed in 28-day mortality rates between the two groups (30% versus 33%), which were similar rates to those observed in our study (25.0% versus 29.9%). In addition, the rate of emerging drug resistance (24.3%) in this study was consistent with our study (25.9%). These similarities between the data presented in our study to those in other studies help support the reliability of our results.

The optimal dosage and duration of meropenem infusion remains elusive [34]. Thus, to determine the optimal meropenem concentration in patients, both its efficacy and safety should be considered. Since drug concentrations at the site of infection influence efficacy, the probability of attaining drug concentrations above the MIC for over 40% of the time between doses (% T > MIC) in the ELF is considered the best marker of bacterial activity for extracellular respiratory tract pathogens [20]. Several studies analyzing the concentration of meropenem on the epithelial lining fluid, according to dosage or duration, showed that a meropenem dosage of 2 g or its prolonged infusion—more than 3 h—could achieve higher exposure in the ELF [21,23]. However, the risk of toxicity—such as neurotoxicity—also increases as the concentration of meropenem increases [35]. A recent study considering both target attainment and potential toxicity reported that the optimal dose regimens were either to administer 2 g every 8 h with 3 h prolonged infusions, or 4 g per day by continuous infusion [36]. Considering these results, administering 1 g of meropenem by 3 h infusion may have provided an insufficient concentration in the ELF against these pathogens—especially those with an MIC > 4 mg/L [21]. Therefore, a randomized, controlled trial of meropenem of 2 g by 3 h infusion or 4 g per day by continuous infusion is needed to evaluate the proper efficacy of extended infusion in patients with pneumonia, including pathogens with an MIC above 4 mg/L.

Our study has several limitations: First, the results of our study are limited by its small sample size and low statistical power, which is a common limitation in retrospective studies. However, it is worth noting that this study included a representative population similar to real-world patients with nosocomial pneumonia. Additionally, due to the retrospective nature of the study, it was not possible to determine the reasons why the medical staff chose a particular method of meropenem infusion. Second, we did not monitor the meropenem antibiotic concentration. Therefore, it is unclear whether the failure to attain target levels by an improper regimen of meropenem caused any differences in clinical outcomes between the two groups. Third, adverse event data were not collected in our study. To optimize the meropenem drug concentration, further studies should investigate its adverse effects—especially in patients who receive high doses of meropenem. Finally, although subgroup analyses may have shown the potential benefits of EI, the results should be interpreted with caution. Large prospective studies are required to confirm the potential advantage of EI over II in these specific subgroups.

## 4. Materials and Methods

### 4.1. Study Design and Patients

We retrospectively collected data from patients treated for nosocomial pneumonia at the medical intensive care unit (ICU) of Asan Medical Center between May 2018 and April 2020, to investigate the efficacy of EI of meropenem compared to intermittent infusion. Eligible patients were aged 18 years or older, were treated for nosocomial pneumonia at the medical ICU, and received meropenem infusion for at least 48 h during their ICU stay. Nosocomial pneumonia was defined as the onset of pneumonia occurring at least 48 h after admission or less than 7 days after discharge from the hospital. Pneumonia occurring within 48 h in patients who stayed for 48 h or longer at another hospital immediately before admission was considered nosocomial pneumonia. Diagnosis of pneumonia was based on clinical assessment by attending physicians, including laboratory and radiologic tests and respiratory signs on electronic medical records. VAP was diagnosed as pneumonia that develops in patients who received mechanical ventilation for over 48 h. Patients were excluded if they did not receive at least 48 h of meropenem treatment in the ICU, had meropenem therapy within 48 h before inclusion, or were simultaneously treated by other beta-lactam antibiotics. 

### 4.2. Infusion of Meropenem

Patients received meropenem by EI (EI group) or intermittent infusion (II group). Patients in the EI group received 1 g of meropenem in 100 mL of normal saline over 3 h of intravenous infusion every 8 h, and immediately after, a loading dose of 1 g in 100 mL of normal saline by intravenous line over 30 min; meanwhile, those in the II group received 1 g meropenem in 100 mL of normal saline by 30 min intravenous infusion every 8 h. Meropenem dosages were adjusted for creatinine clearance across both groups. Drug–drug compatibility was considered during the EI of meropenem. Infusion type, treatment duration, and other antibacterial drugs were determined at the discretion of the medical staff. All patients received standard intensive care from an intensivist throughout their stay in the ICU.

### 4.3. Data Collection and Outcomes

We collected data from electronic medical records, including information on demographics, Sequential Organ Failure Assessment (SOFA) scores, and partial oxygen pressure in arterial blood ((PaO_2_)/fraction of inspired oxygen (FiO_2_) ratio) within 24 h of inclusion. The parameters of the antibiotics, laboratory variables, and organ support at baseline were also recorded. Pathogens from appropriate respiratory specimens collected by bronchoalveolar lavage, endotracheal aspirate, or expectorated sputum and blood samples 48 h before and after the first dose of meropenem were considered as causative species. The pharmacokinetics of meropenem, such as concentrations, were not evaluated.

The primary outcome was a difference in all-cause mortality on day 14 between the two groups. Subgroup analyses for all-cause mortality at day 14 were conducted in patients with empirical treatment and culture-proven Gram-negative pathogens. Patients in the empirical treatment group received meropenem without any previous antibiotics within 48 h or 24 h after initiating antibiotic therapy; they may have also received other antibiotics after starting meropenem, depending on the microbiologic results or the clinical status of the patient. Patients with Gram-negative bacterial pathogens in blood or sputum cultures were considered the culture-proven group. The key secondary outcomes were 28-day mortality, ventilator liberation, and discharge from the ICU on day 14. Other secondary outcomes included clinical response to the improvement of serum C-reactive protein (CRP) levels—which was defined as a decrease in the level at the end of treatment—and the occurrence of carbapenem resistance, which was defined according to Clinical and Laboratory Standards Institute breakpoints (MIC ≥ 4 mg/L for CRPA, ≥4 mg/L for CRAB, and ≥2 mg/L for carbapenem-resistant Enterobacterales (CRE)) [37].

### 4.4. Statistical Analysis

Categorical variables are reported as numbers with percentages, and continuous variables are presented as the mean plus standard deviation (SD) or median with interquartile range (IQR). To compare the variables between the two groups, the Chi-squared test or Fisher’s exact test was used for categorical variables, whereas the Student’s *t*-test or Mann–Whitney U test was used for continuous variables with a normal or non-normal distribution, respectively. The Kaplan–Meier method was used for time-to-event analysis. Time-to-event analysis was right-censored at 14 days. Modeling of the association between time-to-event and variables was estimated by Cox proportional hazards regression analysis. Covariables with statistical differences in comparison between groups or *p*-values < 0.10 in the univariable analysis were selected, considering the collinearity problem. A final model was constructed using a stepwise method for the covariables. The results are presented as hazard ratios (HRs) with 95% confidence intervals (CIs). During the analysis of the ventilator liberation and ICU discharge on day 14, patients who died were assigned 0 days for both. The proportional hazards assumption was assessed through the inspection of Schoenfeld residuals. Two-sided *p*-values of <0.05 were used to indicate significance. All analyses were performed using R software version 4.1.2 accessed on 6 April 2023 (R Core Team).

## 5. Conclusions

In this study, an extended infusion of meropenem for the treatment of nosocomial pneumonia did produce better clinical outcomes than intermittent infusion. However, our results suggest a potential advantage of EI in critically ill, empirically treated or culture-proven nosocomial pneumonia patients with CRE. A large, randomized controlled trial is needed to confirm our findings in specific subgroups.

## Figures and Tables

**Figure 1 antibiotics-12-01542-f001:**
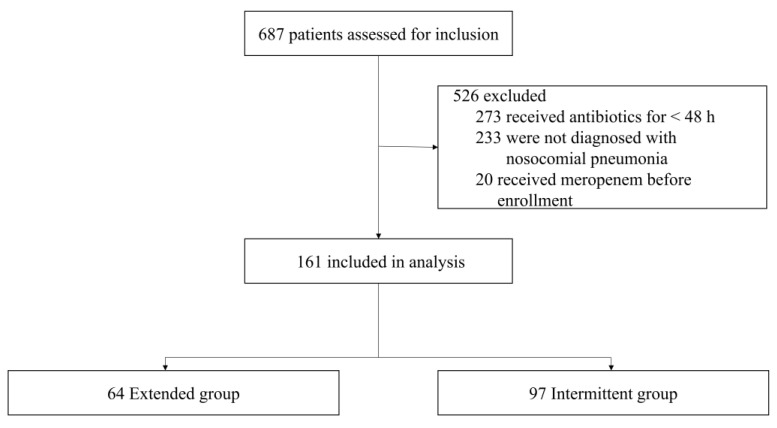
Flowchart of patient population and analysis.

**Figure 2 antibiotics-12-01542-f002:**
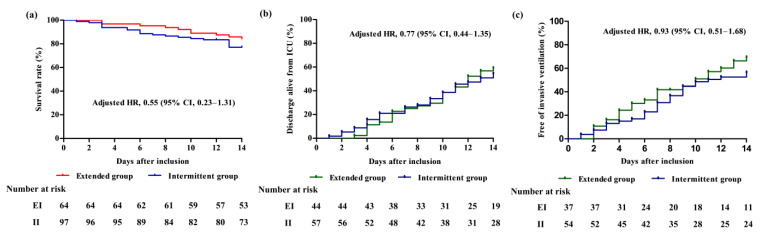
Time-to-event analysis of all patients up to day 14 of inclusion. (**a**) Mortality at 14 days, (**b**) ICU discharge at 14 days, and (**c**) ventilator liberation at 14 days. HR, hazard ratio; CI, confidence interval; EI, extended infusion; II, intermittent infusion.

**Figure 3 antibiotics-12-01542-f003:**
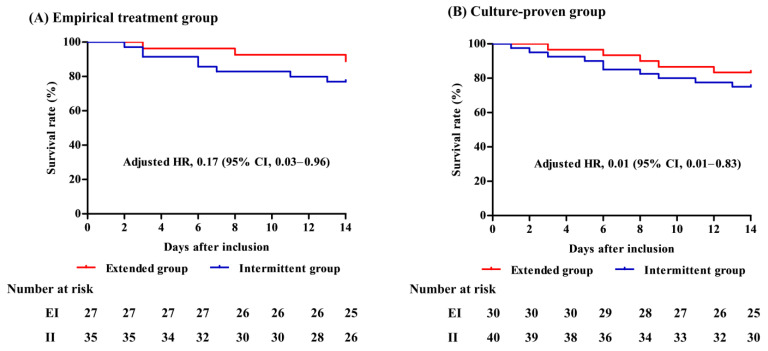
Subgroup mortalities on day 14. (**A**) Empirical treatment group and (**B**) culture-proven group. HR, hazard ratio; CI, confidence interval, EI, extended infusion; II, intermittent infusion.

**Table 1 antibiotics-12-01542-t001:** Baseline characteristics according to the type of meropenem infusion.

Variable	Extended (*n* = 64)	Intermittent (*n* = 97)	*p*-Value
Age, y	62.4 ± 14.1	64.7 ± 12.0	0.255
Sex, male	42 (65.6)	75 (77.3)	0.103
BMI, kg/m^2^	22.9 (19.9–24.8)	22.1 (19.2–24.5)	0.111
**Comorbidity**			
Diabetes mellitus	17 (26.6)	31 (32.0)	0.464
Solid cancer	13 (20.3)	26 (26.8)	0.347
Hematologic malignancy	16 (25.0)	17 (17.5)	0.250
Congestive heart failure	8 (12.5)	5 (5.2)	0.094
Respiratory failure	17 (26.6)	20 (20.6)	0.380
Chronic liver disease	7 (10.9)	14 (14.4)	0.519
End-stage renal disease	9 (14.1)	12 (12.4)	0.755
Immunocompromised	22 (34.4)	25 (25.8)	0.240
**Severity of pneumonia**			
SOFA score	10.0 (6.3–12.0)	10.0 (8.0–13.0)	0.237
Sepsis	61 (95.3)	92 (94.8)	1.000
Septic shock	12 (18.8)	33 (34.0)	0.035
PaO_2_/FiO_2_ ratio, mmHg (*n* = 157)	212.3 ± 98.5	201.2 ± 133.2	0.573
**Type of nosocomial pneumonia**			0.854
Hospital-acquired pneumonia	38 (59.4)	59 (60.8)	
Ventilator-associated pneumonia	26 (40.6)	38 (39.2)	
**Parameters of antibiotics**			
Duration of meropenem, days	9.0 (5.3–13.0)	7.0 (5.0–11.0)	0.165
Duration of antibiotic, days	17.0 (8.3–22.0)	13.0 (7.0–23.5)	0.357
**Discontinuation during ICU stay (*n* = 143)**			0.135
De-escalation or cessation	42 (73.7)	53 (61.6)	
Escalation or death	15 (26.3)	33 (38.4)	
**Previous antibiotics**			
Class of beta-lactam	36 (56.3)	66 (68.0)	0.137
Class of quinolone	18 (28.1)	32 (33.0)	0.514
Class of glycopeptide	11 (17.2)	21 (21.6)	0.487
**Concomitant antibiotics**			
Use of colistin	8 (12.5)	13 (13.4)	0.868
Use of quinolone	24 (37.5)	40 (41.2)	0.742
Use of others *	5 (7.8)	7 (7.2)	1.000
Coverage for MRSA	43 (67.2)	64 (66.0)	0.874
**Laboratory variables**			
White cell count, ×10^3^/L	10.9 ± 8.7	13.2 ± 9.3	0.112
C-reactive protein, mg/dL (*n* = 158)	10.5 ± 8.4	12.4 ± 8.8	0.182
**Organ support at baseline**			
Mechanical ventilation	57 (89.1)	90 (92.8)	0.412
Renal replacement therapy	17 (26.6)	19 (19.6)	0.299
Extracorporeal membrane oxygenation	9 (14.1)	12 (12.4)	0.755
Vasopressors	41 (64.1)	59 (60.8)	0.679

Abbreviations: BMI, body mass index; SOFA, sequential organ failure assessment; PaO_2_, partial oxygen pressure in arterial blood; FiO_2_, fraction of inspired oxygen; MRSA, methicillin-resistant *Staphylococcus aureus*. Data are reported as *n* (%), mean ± standard deviation, or median (interquartile range). * Others included aminoglycoside, tigecycline, ampicillin/sulbactam, cefoperazone/sulbactam, and trimethoprim-sulfamethoxazole.

**Table 2 antibiotics-12-01542-t002:** Microbiological profile of nosocomial pneumonia according to the type of meropenem infusion.

Variable	Extended (*n* = 64)	Intermittent (*n* = 97)	*p*-Value
**Bacteremia**	8 (12.5)	5 (5.2)	0.094
*Staphylococcus aureus*	3 (37.5)	0 (0.0)	
*Enterococcus faecium*	3 (37.5)	0 (0.0)	
*Klebsiella pneumoniae*	2 (25.0)	0 (0.0)	
*Pseudomonas aeruginosa*	0 (0.0)	1 (20.0)	
*Acinetobacter baumanii*	0 (0.0)	4 (80.0)	
**Pathogen of sputum culture**	37 (57.8)	59 (60.8)	0.744
Gram(+) pathogen	4 (6.3)	6 (6.2)	1.000
*Staphylococcus aureus*	3 (75.0)	3 (50.0)	
*Staphylococcus haemolyticus*	1 (25.0)	0 (0.0)	
*Corynebacterium striatum*	0 (0.0)	3 (50.0)	
Gram(−) pathogen	25 (39.1)	45 (46.4)	0.418
*Escherichia coli*	1 (4.0)	1 (2.2)	
*Klebsiella pneumoniae* or *oxytica*	7 (28.0)	10 (22.2)	
*Pseudomonas aeruginosa*	7 (28.0)	11 (24.4)	
*Acinetobacter baumannii or iwoffii*	10 (40.0)	21 (46.7)	
*Stenotrophomonas maltophilia*	1 (4.0)	5 (11.1)	
*Serratia marcescens*	0 (0.0)	1 (2.2)	
CRE	13 (20.3)	28 (28.9)	0.223
CRAB	7 (10.9)	21 (21.6)	0.092
CRPA	6 (9.4)	4 (4.1)	0.197
CPE	0 (0.0)	4 (4.1) *	0.152
**MIC distribution of** **meropenem (*n* = 70)**			0.796
≤1 mg/L	10 (40.0)	15 (33.3)	
2–4 mg/L	2 (8.0)	3 (6.7)	
>4 mg/L	13 (52.0)	27 (60.0)	
**Other pathogens** **	9 (14.1)	6 (6.2)	0.104

Abbreviations: CRE, carbapenem-resistant Enterobacterales; CRAB, carbapenem-resistant *Acinetobacter baumannii*; CRPA, carbapenem-resistant *Pseudomonas aeruginosa*; CPE, carbapenemase-producing Enterobacteriaceae; MIC, minimum inhibitory concentration. * One patient was positive for both CRPA and CPE in the sputum culture. ** Other pathogens indicate virus or fungus as the etiology of pneumonia. Data are reported as *n* (%).

**Table 3 antibiotics-12-01542-t003:** Clinical outcomes according to the type of meropenem infusion.

Outcome Measure	Extended (*n* = 64)	Intermittent (*n* = 97)	*p*-Value
**Mortality**			
14-day mortality	10 (15.6)	22 (22.7)	0.272
28-day mortality	16 (25.0)	29 (29.9)	0.498
ICU mortality	21 (32.8)	40 (41.2)	0.281
**LOS, median (IQR)**			
Hospital days	79.9 ± 76.8	81.3 ± 71.7	0.906
ICU days	26.5 ± 35.6	20.5 ± 18.2	0.223
Ventilator days (*n* = 146)	24.8 ± 37.7	18.5 ± 17.3	0.243
Improvement of CRP level (*n* = 158)	41 (65.1)	61 (64.2)	1.000
Occurrence of carbapenem resistance (*n* = 112)	11 (24.4)	18 (26.9)	0.829

Abbreviations: ICU, intensive care unit; LOS, length of stay; IQR, interquartile range; CRP, C-reactive protein. Data are reported as *n* (%) or mean ± standard deviation.

## Data Availability

Not applicable.

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
