# Peer review of "Extended Versus Intermittent Meropenem Infusion in the Treatment of Nosocomial Pneumonia: A Retrospective Single-Center Study"

_antibiotics, 2023, doi:10.3390/antibiotics12101542_

Round 1

Reviewer 1 Report

This work aimed at comparing differences in terms of major outcomes between EI vs II of meropenem in patients with pneumonia. 

I see some limits.

How clinicians decided to use continuous versus intermitted administration of meropenem? To reduce treatment allocation bias I would suggest to perform a propensity score weighted analysis. 

Table 2 should be more detailed.

Some patients received meropenem even if they had CRAB infection, but it is well known that meropenem does not work on this drug. Were these patients receiving also other drugs? Colistin, tigecyclin, sulbactam?

A better characterization of people who had concomitant bacteremia should be provided (and whether these patients had different characteristics in terms of outcome between the two treatment group). Which agent was responsible? The same which was isolated one respiratory samples?

Any patients with fungal infections?

Minor language issues.

No other comments. 

Minor

Author Response

Reviewer(s)' Comments to Author:

#1 Reviewer’s comments

comment 1) How clinicians decided to use continuous versus intermitted administration of meropenem? To reduce treatment allocation bias I would suggest to perform a propensity score weighted analysis.

Response: The attending staff decided on the type of meropenem administration. The reason for this choice could not be determined due to the retrospective nature of the study. We added this information to the Materials and Methods section (Page 8, Materials and Methods, Line 244) and the Discussion section (Page 7, Discussion, Lines 209–211). We performed propensity-score matching to balance the baseline characteristics, such as age, sex, and SOFA score. The results from the propensity-score matching are shown in the tables and figures below. It was difficult to conduct propensity-score matching due to the limited number of patients and the diversity in the underlying conditions, causative pathogens, and antibiotic co-usage. The power of the results did not present any statistical significance. We have added these tables and figures to the Supplementary files.

comment 2) Table 2 should be more detailed.

Response: We have modified Table 2 to provide more detailed microbiological information.

comment 3) Some patients received meropenem even if they had CRAB infection, but it is well known that meropenem does not work on this drug. Were these patients receiving also other drugs? Colistin, tigecyclin, sulbactam?

Response: Twenty-eight patients with a CRAB infection received meropenem treatment when the microbiological test of antibiotic susceptibility was unknown. After the confirmation of antibiotic susceptibility, 16 (57.1%) of the 28 patients received high doses (9 g over 4 h) of ampicillin/sulbactam, cefoperazone/sulbactam, tigecycline, or amikacin. An additional 8 patients (28.7%) continued meropenem or changed from meropenem to other beta-lactam antibiotics because of an improvement in their medical status. Only 4 (14.3%) patients died while using meropenem.

comment 4) A better characterization of people who had concomitant bacteremia should be provided (and whether these patients had different characteristics in terms of outcome between the two treatment group). Which agent was responsible? The same which was isolated one respiratory samples?

Response: We have produced and provided a table and figure to illustrate the characteristics of the patients and the comparison of the 14-day mortality between the extended infusion group and intermittent infusion group in patients with concurrent bacteremia. We have added these to the Supplementary files. In addition, we have added more detailed microbiological information to Table 2 and the Results section (Page 4, Results, Lines 90–101).

comment 5) Any patients with fungal infections?

Response: As mentioned in Table 2, there were 15 patients with a viral or fungal pathogen with an etiology of pneumonia, of which five patients had a fungal infection (aspergillosis fumigatus). One patient died in the early stages of the meropenem treatment, while after meropenem treatment, four patients received non-carbapenem beta-lactam antibiotics with other drugs, such as fluconazole, voriconazole, or micafungin.

comment 6) Minor language issues.

Response: A professional proofreading company has reviewed and corrected any language issues found in the manuscript. We have uploaded the English proofreading certification to confirm this process.

Reviewer 2 Report

The aims and methods of this study were clearly described. However, there are some suggestion / questions to improve this manuscript.

Firstly, although there is no recommended structure of manuscript writing, I would suggest the authors to rearrange the structure of manuscript by moving the methods section to topic 2. This will make the manuscript easier to read.

Secondly, it is surprising to see the similarity between the extended infusion and intermittent groups in all baseline characteristics, even though this was retrospective study and nothing was controlled. Was any method used for categorising patients into each group to balance their characteristics?

In topic 2.3, it would be better to divide subgroup analysis results to another paragraph.

In table 1, please make the topics of each characteristic (e.g. "Severity of pneumonia" and "Parameters of antibiotics") bold or italic to make them easy to read.

In topic 4.3, please explain more about "empirical treatment". What were the antibiotics that the patients received before meropenem, and were they continuously used after the patients received meropenem?

In discussion section, the authors stated "1 g by 3 h infusion of meropenem in our study 172 may have resulted in an insufficient concentration in the ELF...". It would be better to compare the results in this study, such as mortality rate, to the results from previous studies which used the same dose. Perhaps the survival rates of this dose were acceptable, whether or not extended of intermittent infusion.

Also, it would be more interesting to show the results or discuss the possibility to use extended versus intermittent infusion in specific patients, such as the patients who were infected with non-carbapenem resistant bacteria because the figure 3(C) shows extended infusion group had better survival rate than the intermittent group.

This study was well written. No major English mistake was detected.

Author Response

#2 Reviewer’s comments

comment 1) Firstly, although there is no recommended structure of manuscript writing, I would suggest the authors to rearrange the structure of manuscript by moving the methods section to topic 2. This will make the manuscript easier to read.

Response: We appreciate your recommendation; however, the instructions for Authors for the journal of ‘Antibiotics (Basel)’ recommends placing the Methods section in topic 4. Therefore, we have maintained the current structure of the manuscript.

comment 2) Secondly, it is surprising to see the similarity between the extended infusion and intermittent groups in all baseline characteristics, even though this was retrospective study and nothing was controlled. Was any method used for categorising patients into each group to balance their characteristics?

Response: No special method was used to categorize patients and balance their characteristics. The use of the same admission criteria alongside the consistent care by the same intensivists throughout the study period and shared protocols within the medical intensive care unit may be associated with this balance.

comment 3) In topic 2.3, it would be better to divide subgroup analysis results to another paragraph.

Response: We have divided the results of the subgroup analysis into another paragraph as per your recommendation (Page 5, Results, Lines 126–135).

comment 4) In table 1, please make the topics of each characteristic (e.g. "Severity of pneumonia" and "Parameters of antibiotics") bold or italic to make them easy to read.

Response: We have changed the topics of each characteristic into bold and aligned them to the center to improve the readability, as per your recommendation.

comment 5) In topic 4.3, please explain more about "empirical treatment". What were the antibiotics that the patients received before meropenem, and were they continuously used after the patients received meropenem?

Response: We have defined the subgroup of patients receiving empirical treatment as those who received meropenem without previous antibiotics within 48 h or 24 h after initiating antibiotic therapy. These patients received meropenem after discontinuing any previous beta-lactam antibiotic treatments, although subsequent changes in antibiotic treatment may have occurred based on microbiological results or patient status. We have added this confusion and have provided a more detailed explanation of the subgroup. (Page 8–9, Materials and Methods, Lines 261–263).

comment 6) In discussion section, the authors stated "1 g by 3 h infusion of meropenem in our study 172 may have resulted in an insufficient concentration in the ELF...". It would be better to compare the results in this study, such as mortality rate, to the results from previous studies which used the same dose. Perhaps the survival rates of this dose were acceptable, whether or not extended of intermittent infusion.

Response: Considering the results in references 27, 32, and 33, including patients who received 1 g of meropenem every 8 h for nosocomial pneumonia or sepsis, with the same dose as in our study, the mortality rate in our study (28-day mortality: 25.0% in the extended infusion group and 29.9% in the intermittent infusion group) was not inferior to those in other studies, indicating the dose of meropenem in our study is a generally acceptable dose in real-world medical treatments. However, this dose of meropenem may not provide sufficient concentrations in the ELF for the pathogen of MIC > 4 mg/L. Therefore, we have modified the sentence in the Discussion section to clarify the information being provided (Page 7, Discussion, Lines 173–187).

comment 7) Also, it would be more interesting to show the results or discuss the possibility to use extended versus intermittent infusion in specific patients, such as the patients who were infected with non-carbapenem resistant bacteria because the figure 3(C) shows extended infusion group had better survival rate than the intermittent group.

Response: We have added the Kaplan–Meier curve for patients with non-carbapenem-resistant bacteria (Supplementary Figure 1). In patients with non-CRE, there was no difference in the 14-day mortality between the two groups (25.0% in the extended infusion group versus 17.6% in the intermittent infusion group, p = 0.554), while the extended infusion group (0.0%) tended to have lower 14-day mortality rates compared with the intermittent infusion group (25.0%, p = 0.056) in patients with CRE (Page 5, Results, Lines 132–135)

Reviewer 3 Report

In your manuscript, you present a retrospective study from a single center that assesses the two dosage regimens of meropenem currently under debate. The article addresses an interesting topic; however, the limitations outlined in the manuscript significantly undermine the validity of the results. It is not possible to compare the two dosage regimens without measuring the lung concentrations in the patients, especially given their critical condition. Furthermore, considering the 13% incidence of renal insufficiency and 23% of patients undergoing dialysis, achieving optimal antibiotic concentration becomes even more challenging.

Allow me to provide you with a series of recommendations that can enhance the quality of your manuscript:

  1. Incorporate the severity of infection for each patient, distinguishing between sepsis, severe sepsis, and septic shock. This variable holds significant importance and can serve as a confounding factor in uncontrolled Cox models.
  2. Introduce 28-day mortality as a secondary outcome.
  3. Present the calculation of the sample size or the potential beta error that might be present.
  4. Elaborate in greater detail, within the Materials and Methods section, on the construction of the Cox regression models.
  5. Consider including a table featuring the final models, potentially as supplementary material.
  6. In Figures 2 and 3, clarify whether Kaplan-Meier models or Cox regression were employed, and also include the number of individuals at risk in each case.

Your attention to these recommendations will contribute to the refinement of your manuscript's content

I recommend a review of the manuscript, as there are certain sentences that are challenging to comprehend. For instance, in the introduction, lines 46 to 48 pose difficulties in understanding.

Author Response

#3 Reviewer’s comments

comment 1) Allow me to provide you with a series of recommendations that can enhance the quality of your manuscript:

Response: Thank you very much for the careful review and insightful comments on our manuscript.

comment 2) Incorporate the severity of infection for each patient, distinguishing between sepsis, severe sepsis, and septic shock. This variable holds significant importance and can serve as a confounding factor in uncontrolled Cox models.

Response: We have added the variables relating to sepsis and septic shock to the severity of pneumonia section in Table 1 and conducted a new analysis using the multivariable Cox proportional hazards model, which now includes sepsis and septic shock as covariates. Despite adjusting for sepsis and septic shock, no changes were observed in the clinical outcomes. We have updated the relevant sentences alongside uploading a new Supplementary file, which includes the tables with the results from the multivariate models (Page 5, Results, Lines 117–121).

comment 3) Introduce 28-day mortality as a secondary outcome.

Response: We have introduced the 28-day mortality rate as a secondary outcome (Page 9, Materials and Methods, Line 265) and included the result in Table 3. Similar to the other outcomes, no difference was observed in the 28-day mortality rate between the two groups (25.0% in the extended infusion group versus 29.9% in the intermittent infusion group, p = 0.498).

comment 4) Present the calculation of the sample size or the potential beta error that might be present.

Response: When calculating the power of this study, assuming a 1:1 ratio in the study group, a 30% in-hospital mortality rate was observed in the II group, along with a 50% reduction in the in-hospital mortality rate in the EI group, and an alpha value of 0.05 based on previous studies (reference 15, 26-28). This study demonstrated a low power of approximately 65%, which is characteristic of many retrospective studies. We have included the low power of our study as a limitation in the Discussion section (Page 7, Discussion, Lines 206–209).

comment 5) Elaborate in greater detail, within the Materials and Methods section, on the construction of the Cox regression models.

Response: We have added a sentence to elaborate on the details of the Cox regression models (Page 9, Materials and Methods, Line 282) and have updated the tables to include the multivariate models as Supplementary files.

comment 6) Consider including a table featuring the final models, potentially as supplementary material.

Response: As mentioned above, we have added the new Supplementary files, which include the tables for the multivariate models.

comment 7) In Figures 2 and 3, clarify whether Kaplan-Meier models or Cox regression were employed, and also include the number of individuals at risk in each case.

Response: We have corrected Figures 2 and 3 to include the number of individuals at risk Table and have added further detailed information from the multivariate Cox regression analysis into the Materials and Methods section (Page 9, Materials and Methods, Lines 282).

comment 8) Your attention to these recommendations will contribute to the refinement of your manuscript's content

Response: As per your recommendation, we have revised the manuscript to enhance its quality.

comment 9) I recommend a review of the manuscript, as there are certain sentences that are challenging to comprehend. For instance, in the introduction, lines 46 to 48 pose difficulties in understanding.

Response: A professional proofreading company has reviewed and corrected any language issues found in the manuscript, including the sentences from lines 46 to 48 (Page 2, Introduction, Lines 46–49). Moreover, we have uploaded the English proofreading certification to confirm this process.

Round 2

Reviewer 3 Report

Dear authors

The substantial endeavor undertaken by the authors to improve the manuscript is acknowledged. However, a study lacking adequate statistical power can lead to erroneous conclusions due to inaccurate sample size estimation. The minimum required power for this study should be 80%, especially given its retrospective nature and the absence of unbiased group selection.

My recommendation is to identify and continue to include patients from the last two years (2021 and 2022), which will undoubtedly increase the study's sample size.

Please be aware that your conclusions suggest that continuous meropenem infusion does not contribute to 14- or 28-day outcomes, based on a clearly insufficient sample size. Furthermore, if you reduce the sample size even further in a subgroup analysis, thus diminishing statistical power, it appears to favor continuous infusion in terms of 14d mortality.

Sending contradictory messages to the scientific community may occur if inappropriate messages are disseminated based on methodologically weak studies.

          Best Regards